

# Early detection of cyanide, organophosphate and rodenticide pollution based on locomotor activity of zebrafish larvae

Binjie Wang[1], Junhao Zhu[1], Anli Wang[1,2], Jiye Wang[1], Yuanzhao Wu[1] and Weixuan Yao[1]

[1] The Department of Criminal Science and Technology, Zhejiang Police College, Key Laboratory of Drug Prevention and Control Technology of Zhejiang Province, Hangzhou, Zhejiang province, People's Republic of China

[2] College of Biosystems Engineering and Food Science, Zhejiang University, National Engineering Laboratory of Intelligent Food Technology and Equipment, Zhejiang Key Laboratory for Agro-Food Processing, Fuli Institute of Food Science, College of Biosystems Engineering and Food Science, Hangzhou, Zhejiang Province, People's Republic of China

Corresponding authors
Binjie Wang, wangbinjie@zjjcxy.cn
Weixuan Yao, yaoweixuan@zjjcxy.cn

## ABSTRACT

Cyanide, organophosphate and rodenticides are highly toxic substances widely used in agriculture and industry. These toxicants are neuro- and organotoxic to mammals at low concentrations, thus early detection of these chemicals in the aqueous environment is of utmost importance. Here, we employed the behavioral toxicity test with wildtype zebrafish larvae to determine sublethal concentrations of the above mentioned common environmental pollutants. After optimizing the test with cyanide, nine rodenticides and an organophosphate were successfully tested. The compounds dose-dependently initially (0–60-min exposure) stimulated locomotor activity of larvae but induced toxicity and reduced swimming during 60–120-min exposure. $IC_{50}$ values calculated based on swimming distance after 2-h exposure, were between 0.1 and 10 mg/L for both first-generation and second-generation anticoagulant rodenticides. Three behavioral characteristics, including total distance travelled, sinuosity and burst count, were quantitatively analyzed and compared by hierarchical clustering of the effects measured by each three parameters. The toxicity results for all three behavioral endpoints were consistent, suggesting that the directly measured parameter of cumulative swimming distance could be used as a promising biomarker for the aquatic contamination. The optimized method herein showed the potential for utilization as part of a monitoring system and an ideal tool for the risk assessment of drinking water in the military and public safety.

Subjects Animal Behavior, Aquaculture, Fisheries and Fish Science, Biochemistry, Toxicology, Ecotoxicology
Keywords Aquatic contamination, Rodenticide, Danio rerio, Monitoring system, Early warning

## INTRODUCTION

With the intensification of anthropogenic activities, the pressure for routine surveillance to manage drinking water and environmental water quality is growing for public health authorities (*Van der Schalie et al., 2001*; *Nath et al., 2013*). Low concentrations of toxic substances at the ng/L level, such as cyanide, organophosphates and rodenticides, have

been present in the aqueous environment (*Gomez-Canela, Barata & Lacorte, 2014*). Higher concentrations of such chemicals in aqueous solutions can cause acute poisoning in humans, as reported in human poisoning incidents and public safety emergencies (*Palmer et al., 1999*; *Olmos & Lopez, 2007*; *Hamel, 2011*; *Wang et al., 2016*). The toxic mechanisms of these substances to mammals have been well studied. Specifically, cyanide is a broad-spectrum poison as a potent cytochrome C oxidase inhibitor and causes toxic hypoxia in tissues (*Sabourin et al., 2016*). Organophosphates, such as parathion, malathion and chlorpyrifos, covalently inhibit acetylcholinesterase (AChE) and prevent the breakdown of the neurotransmitter acetylcholine, which can cause seizures, cardiovascular and respiratory failure, and death in humans at high doses of exposure (*Garcia et al., 2003*). Most rodenticides, such as diphcinone, flocoumafen, coumachlor, chlorophacinone, coumatetralyl, difenacoum, brodifacoum and bromadiolone, are vitamin K antagonists that block the formation of prothrombin, inhibit normal blood clotting and are highly toxic to mammals (*Regnery et al., 2018*). There is still a possibility of acute poisoning of humans with high concentrations of toxicants, such as accidental water contamination, poisoning cases and other potential public safety hazards in incidents or even war. Compared with the established instrumental detection methods applied to known toxicants, a fast and effective early warning method is lacking for unknown toxicants in water at high concentrations, and behavioral experiments with model animals can fill this gap.

Zebrafish (Danio rerio) is one of the most used species in the field of toxicology, pharmacology and pharmacogenomic studies (*De Esch et al., 2012*). With the advantages of low cost, high productivity and rapid reproduction, zebrafish have been promising laboratory models for assessing behavioral toxicity (*Storey, Gaag & Burns, 2011*; *Bae & Park, 2014*). Importantly, the existing studies correlated behavioral changes of zebrafish induced by toxicant exposure with alterations in physiological indicators, thus providing an established basis for simple and rapid water quality toxicity evaluation (*Zhang et al., 2016*). Several behavioral parameters in zebrafish, such as locomotion, avoidance, aggression, memory and others (*Scott & Sloman, 2004*; *Blaser & Gerlai, 2006*) have been used as toxicological endpoints easily observed and quantified in the biological early warning system (BEWS). For example, by the continuous recording of the physiological response of organisms in aqueous solutions, it was possible to rapidly analyze the behavioral alterations caused by adverse biological effects at sub-lethal levels of the contaminants (*Storey, Gaag & Burns, 2011*; *Li et al., 2019*). Behavioral responses are faster and 10-1000 times more sensitive compared to other conventional endpoints such as lethality (*Huang et al., 2010*). Especially for chemically induced stress, behavioral changes of zebrafish are more sensitive and faster indicators than the morphological criteria traditionally used for ecotoxicological tests. Rapid technology improvement in video-tracking devices and compatible software has facilitated the application of quantitative analysis of behavior in toxicity evaluation.

Adult zebrafish have been employed as indicators of water contamination by measuring their behavioral abnormalities (*Scott & Sloman, 2004*; *Geng & Peterson, 2019*). For example, through analyzing angular velocity, linear velocity, spatial dispersion and other behavioral components, (*Oliva Teles et al., 2015*) developed a promising probabilistic neural network (PNN) statistical model to detect commercial bleach at 0.0005% (v/v) which corresponded

to 0.5% of the 24-h $LC_{50}$ value of NaOCl for adult zebrafish. Recently, changes of advanced behaviors, such as courtship behavior (*Li et al., 2019*), exploratory activity (*Amorim et al., 2018*), and predator avoidance (*Amorim et al., 2017*) were also reported in detection of chemical contamination.

In addition, zebrafish larval behavior has been used for the early warning of contamination, with the advantages of high throughput, high sensitivity, low cost and easily measurable behavioral endpoints (*Colwill & Creton, 2011*; *Nusser et al., 2016*). For example, *Carbaugh et al. (2020)* reported the use of photomotor response of zebrafish embryos at 24 h post fertilization (hpf) for toxicity screening of cyanide, organophosphorus pesticides and chemical weapon precursor compounds. The authors showed that the visual neuronal pathways were fully established in zebrafish larvae at 96 hpf, and could elicit significant abnormalities in startle latencies within minutes of sublethal cyanide exposure. In another study, exposure to insecticides at a concentration of 1 ppb in the aquatic environment significantly altered three endpoints of zebrafish larval behavior, including total distance traveled, bursts, and number of rotations. However, the authors also mentioned that pesticides triggered different levels of locomotor responses in zebrafish larvae, possibly due to some toxicants exerting stimulatory, instead of damaging, effects on zebrafish larvae at low concentrations (*Hussain et al., 2020*). Other behavioral parameters , such as thigmotaxis, avoidance behavior, resting time and swimming speed have also been used for behavioral analysis after exposure to environmental toxicants or drugs (*Richendrfer & Creton, 2013*; *Richendrfer & Creton, 2018*).

The studies so far have focused on one or two types of toxicants, whereas the early warning methods must be able to work with multiple types of toxicants with different toxicity mechanisms. The aim of this work was to develop a robust system with sensitive detection capability of several toxicants in water, based on the behavioral changes of zebrafish larvae. We hypothesized that the zebrafish behavioral test would be suitable for use as a sensitive detection method for the set of toxic chemicals included in the study. To test this hypothesis, first, a series of potassium cyanide (KCN) exposure tests on zebrafish larvae at 6 days post fertilization (dpf) were conducted to establish an assessment method that revealed the relationship between toxicant concentration and behavioral toxicity, and these tests were further validated on organophosphate insecticide (methyl parathion), insecticide and rodenticide (fluoroacetamide), and anticoagulant rodenticides (diphcinone, flocoumafen, coumachlor, chlorophacinone, coumatetralyl, difenacoum, brodifacoum, and bromadiolone). These selected environmental toxicants act *via* different mechanisms of toxicity, but all induced a significant reduction in the swimming distance of the zebrafish exposed to the toxicants at concentrations of 1 or 10 mg/L. The method enabled to correlate the toxic response in zebrafish larvae with high concentrations of water contamination, showing the potential of the proposed method as part of a monitoring system in the military and public safety.
## MATERIALS & METHODS

### Zebrafish maintenance and embryo collection

The adult wildtype zebrafish (∼3 months old) were purchased from China Zebrafish Resource Center (CZRC, Wuhan, China). They were fed twice a day with brine shrimp and maintained at 28 °C with a 14-h light/10-h dark photoperiod (lights on at 8:00 a.m.). This work utilized 144 h post fertilization (hpf) zebrafish larvae hatched from healthy eggs. For the first 5 days of development, larvae were kept at 28 °C in a light: dark rhythm of 14:10 h. The selected fish embryos were checked daily, and dead and malformed embryos were removed. The zebrafish larvae were not fed during the experiment. At 6 dpf, the healthy zebrafish larvae were exposed to different concentrations of chemicals and assessed for behavioral changes for 120 min. The zebrafish experiments were approved by the Zhejiang University Experimental Animal Welfare and Ethical Review Committee (Ethical approval number: ZJU20210168).

### Toxicants

A standard solution containing KCN was purchased from O2si Smart Solutions (Charleston, SC, USA), and methyl parathion from Accu Standard (Connecticut, USA). Chlorophacinone, difenacoum, bromadiolone, diphcinone, brodifacoum, coumatetralyl, flocoumafen, fluoroacetamide and coumachlor were purchased as powders from AChemTek, Inc. (Worcester, MA, USA). The stock solutions of KCN, difenacoum, bromadiolone, diphcinone, coumatetralyl, flocoumafen, and fluoroacetamide were prepared in deionized $H_2O$ and stored in the dark at 4 °C for up to 24 h. Stock solutions of chlorophacinone, brodifacoum and coumachlor were prepared in DMSO, and the final concentration of DMSO in the exposure solutions was 0.1% (*Christou et al., 2020*). All the toxicants were tested at five different concentrations (0, 0.01, 0.1, 1 and 10 mg/L), prepared as dilutions of the stock solutions with formulated water.

### Chemical exposures

The experiments were conducted following published protocols (*Nusser et al., 2016*). Normally developed larvae were collected at 6 dpf in a volume of 270 μL formulated water and randomly transferred into an individual well of a clear polystyrene flat-bottom 96-well plate between 9:00 am and 5:00 pm. To begin the exposures, 30 μL of 10 times concentrated toxicant solution was added to the well containing 270 μL of formulated water with a zebrafish larva. Each concentration of the contaminant was tested in 12 replicates, *i.e.,* 12 wells, each containing one larva. A total of 660 larvae were used.

### Behavioral experiments

Larvae behavior was recorded using the DanioVisionTM observation system (Noldus, Wageningen, Netherlands). After 10 min of acclimation in the dark, we recorded behavioral changes of zebrafish larvae during 120 min, including six alternating cycles of 10 min of light phase and 10 min of dark phase (infrared light). The EthoVision XT 10 software package (Noldus Information Technology, Leesburg, VA, USA) was used to analyze the generated video data. The data were smoothed and the sampling point was set to the

previous position until the movement distance was greater than 0.20 mm. Finally, the absence of heartbeat observed by the stereomicroscope (Olympus, SZX2)was used as an indicator of death.

The accumulated behavioral data were collected in each 60-minute period, consisting of 3 periods of light cycles and 3 periods of dark cycles. The activity of each zebrafish larvae was analyzed for three endpoints, namely, total distance traveled (mm), sinuosity (°/s) and burst count, based on previous reports (*Hussain et al., 2020*; *Wang et al., 2020*). Briefly, sinuosity was defined as the deflection angle divided by the distance traveled, and was associated with neurodevelopmental toxicity. For the burst activity count, the quantization of video track parameters was set as burst (20), freeze (2), and detection threshold (20) (*Wang et al., 2020*). The inhibitory capacity of the toxicant on the behaviour of zebrafish was assessed as the $IC_{50}$, *i.e.,* the concentration at which the cumulative distance travelled was reduced by 50% over 60–120 min compared to the control group (*Broening et al., 2019*).

### Statistical analysis

A one-way analysis of variance (ANOVA) was used to examine the differences in motor behavior between the treatment and control group (0 mg/L). In case data set failed the normality and homoscedasticity test, Welch's ANOVA test was performed. When differences were significant, each treatment was further compared with control using appropriate post hoc test (Dunnett's T3 multiple comparisons test). Prism 9 (GraphPad, US) was used to analyze significant differences between the exposed and control groups. Statistical significance threshold was set at $^*p < 0.05$, $^{**}p < 0.01$, $^{***}p < 0.001$ and $^{***}p < 0.001$ for all experiments. Heatmap was plotted using the Omic Share tools, a free online platform for data analysis (http://www.omicshare.com/tools).

## RESULTS

### Optimization of the zebrafish behavioral toxicity test using KCN

The larval zebrafish (6 dpf) were exposed to different concentrations of KCN (0, 0.01, 0.1, 1, 10 mg/L). We recorded behavioral data from the beginning of the exposure for 2 h and then ended the exposure experiment. Behavioral data recorded by the DanioVision TM observation system were exported to Prism software for further analysis. During each light/dark transition period (20 min), the larvae moved in cycles which consisted of shorter distances during 10-min light periods and longer distances during 10-min dark periods (Fig. 1A). At 10 mg/L, mortality of zebrafish larvae caused the failure of meaningful behavioral analyses.

The zebrafish gradually adapted to the light stimulus, which resulted in a gradual decrease in motility. In particular, the controls did not show a consistent decline in each cycle of light and dark changes, which made it difficult to assess mobility caused by toxic exposure.Based on preliminary experiments, 3 cycles of light and dark were necessary to observe effects with all tested chemicals. Then, we assessed the behavioral impairment of zebrafish using the cumulative distances travelled by zebrafish larvae over the three cycles totaling the first 60 min (0–60 min) and over three cycles of the second 60 min (60–120 min) of exposure.

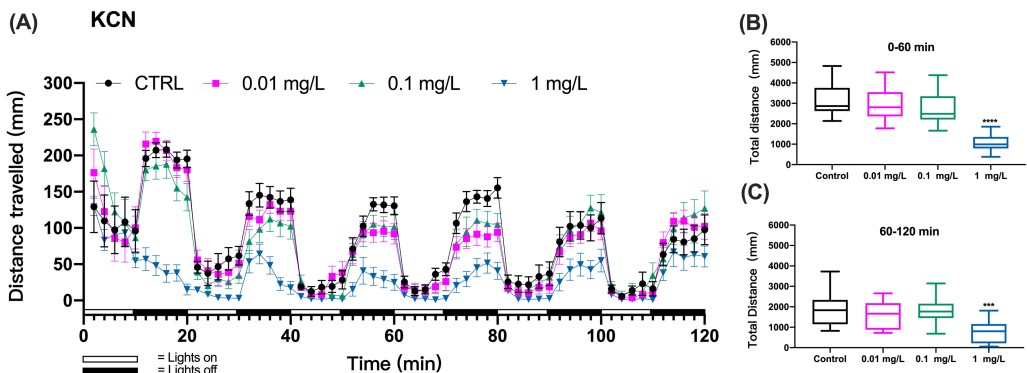

**Figure 1  KCN exposure induced inhibition of locomotor activity in zebrafish larvae.** (A) Distance travelled for each 2 min during the KCN exposure. (B) Total distance travelled for the first 60 min during the KCN exposure. (C) Total distance travelled for the second 60 min during the KCN exposure. Data are presented as boxplots with the median (12 larvae per treatment). An asterisk (*) indicates the concentrations which induced statistically significant difference compared to control (negative control). Significance was defined as *** $p < 0.001$, **** $p < 0.0001$.

The results showed that the KCN exposure at 1 mg/L resulted in significant reduction in the cumulative swimming distance compared to the control group both during the first hour (one-way ANOVA, $F_{(3, 44)} = 21.82$, $p < 0.0001$ $p < 0.0001$; Dunnett's T3 post hoc test, $p < 0.0001$) and the second hour of exposure (one-way ANOVA, $F_{(3, 44)} = 6.633$, $p = 0.0009$; Dunnett's T3 post hoc test, $p = 0.0008$) in the 1 mg/L KCN exposure group (Figs. 1B and 1C). Since 10 mg/L KCN was apparently lethal, there was no swimming detected in this exposure group.

## Corroboration of the behavioral toxicity assay with insecticides and rodenticides

The method was further applied to detect the behavioral changes in zebrafish larvae exposed to organophosphate insecticide (methyl parathion), insecticide and rodenticide (fluoroacetamide), or anticoagulant rodenticides (diphcinone, flocoumafen, coumachlor, chlorophacinone, coumatetralyl, difenacoum, brodifacoum, and bromadiolone). Zebrafish larvae exposure to the first-generation anticoagulant rodenticide coumachlor produced a similar cyclic swimming pattern in alternating light and dark cycles as observed in the test with KCN. However, differently from KCN, 1 and 10 mg/L coumachlor induced clearly different swimming distances from the control larvae. During the first, second and third cycles the larvae exposed to 1 and 10 mg/L coumachlor swam significantly longer distances than control larvae (one-way ANOVA, $F_{(5, 65)} = 10.33$, $p < 0.0001$; Dunnett's T3 post hoc test, $p = 0.0024$ and $p = 0.0042$, respectively). This pattern disappeared during the next light cycles (60–120 min), however, the swimming distance was significantly shorter in 10 mg/L coumachlor group compared to control (one-way ANOVA, $F_{(5, 63)} = 13.01$, $p < 0.0001$; Dunnett's T3 post hoc test $p < 0.0001$), in the second exposure hour. In addition, there was no significant difference between the cumulative movement distance of zebrafish larvae in the solvent control group (0.1% DMSO) compared with the negative

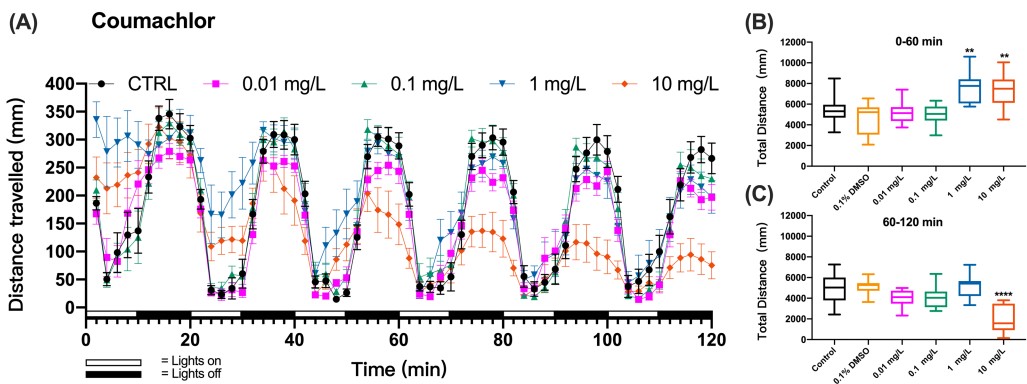

**Figure 2** **Coumachlor exposure induced alteration of locomotor activity in zebrafish larvae.** (A) Distance travelled for each 2 min during the coumachlor exposure. (B) Total distance travelled for the first 60 min during the coumachlor exposure. (C) Total distance travelled for the second 60 min during the coumachlor exposure. Data are presented as boxplots with the median (12 larvae per treatment). An asterisk (*) indicates the concentrations which induced statistically significant difference compared to control (negative control). Significance was defined as ** $p < 0.01$, **** $p < 0.0001$.

control group within 60–120 min (Dunnett's T3 post hoc test, $p = 0.9997$). These results were well illustrated by the analysis of the cumulative distances travelled by zebrafish larvae over the first 60 min (0–60 min) and second 60 min (60–120 min) of exposure (Figs. 2B and 2C), indicating a time delay in the behavioral effects in fish.

## Corroboration of the behavioral toxicity assay with insecticides and rodenticides

When zebrafish larvae were exposed to a second-generation anticoagulant brodifacoum, the pattern of swimming distances over time that emerged at different toxicant concentrations differed from these of KCN and coumachlor (Fig. 3A). Based on the time course of the swimming distance, there were no significant differences between the control group and 0.01, 0.1, 1 and 10 mg/L brodifacoum groups during the "lights on" and "lights off" cycles (0–60 min) (one-way ANOVA, $F (5, 66) = 1.859$, $p = 0.1136$). However, 10 mg/L brodifacoum induced longer travel distances, compared to the control larvae, starting from the initiation of toxicant exposure until the second "lights on" cycle (20–30 min). After that, the swimming distance in 10 mg/L brodifacoum sharply decreased and continued to decline until after 90 min of exposure there was no movement of fish detected. The time-dependent toxicity of brodifacoum was also illustrated in the comparison of the cumulative distances traveled during the first and second exposure hour: there were no changes in swimming distances compared to untreated control during the first hour of exposure (Fig. 3B), but, similarly with coumachlor, also brodifacoum at 10 mg/L induced significant inhibition in the cumulative swimming distance during the second hour of exposure (Fig. 3C) (Welch's ANOVA test, $F (5.000, 28.03) = 53.30$, $p < 0.0001$; Dunnett's T3 post hoc test, $p < 0.0001$), suggesting that a longer exposure was needed for the onset of behavioral effects. No significant difference between the cumulative movement distance

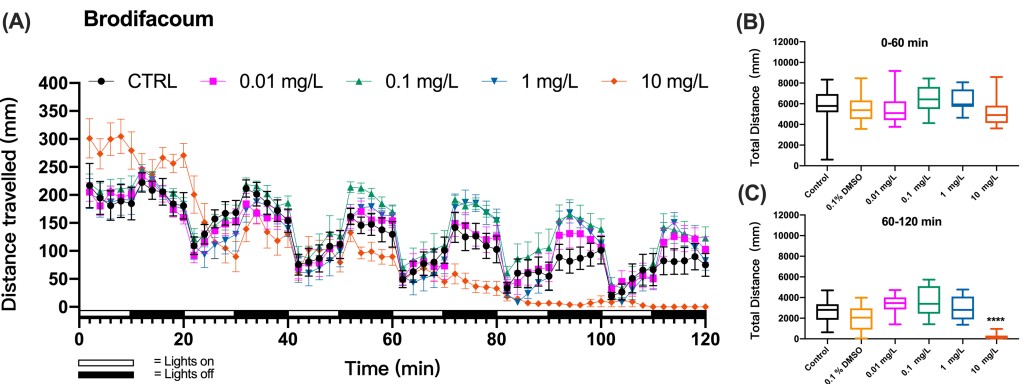

**Figure 3 Brodifacoum exposure induced alteration of locomotor activity in zebrafish larvae.** (A) Distance travelled for each 2 min during the brodifacoum exposure. (B) Total distance travelled for the first 60 min during the brodifacoum exposure. (C) Total distance travelled for the second 60 min during the brodifacoum exposure. Data are presented as boxplots with the median (12 larvae per treatment). An asterisk (*) indicates the concentrations which induced statistically significant difference compared to control (negative control). Significance was defined as **** $p < 0.0001$.

of zebrafish larvae in the solvent control group (0.1% DMSO) compared with the negative control group within 60–120 min was found (Dunnetts T3 post hoc test, $p = 0.5738$).

The behavioral changes in zebrafish larvae were also measured during the exposure to organophosphate insecticide (methyl parathion), insecticide and rodenticide (fluoroacetamide), and anticoagulant rodenticides (diphcinone, flocoumafen, chlorophacinone, coumatetralyl, difenacoum, and bromadiolone) at different concentrations. Since the results of swimming distance changes over time indicated that the toxicity of these chemicals was time-dependent, manifesting the behavioral toxicity at the second hour of exposure, similar to the effects of coumachlor and brodifacoum (Figs. 2A and 3A, respectively), the effects of other chemicals were compared based on the total distance travelled by zebrafish larvae during the second hour of exposure (Fig. 4). For all the toxicants tested, the behavioral abilities of larvae at concentrations of 0.01 mg/L and 0.1 mg/L were not significantly different compared with the control group. At the concentration of 1 mg/L, chlorophacinone caused significant decrease in the mobility of zebrafish (Fig. 4C) (Welch's ANOVA test, F (4.000, 22.81) = 48.43, $p < 0.0001$; Dunnett's T3 post hoc test, $p = 0.0002$). For methyl parathion (one-way ANOVA, F (4, 55) = 7.986, $p < 0.0001$; Dunnett's T3 post hoc test, $p = 0.0131$) and coumatetralyl (one-way ANOVA, F (4, 55) = 18.07, $p < 0.0001$; Dunnett's T3 post hoc test, $p = 0.0159$), 1 mg/L exposure concentration also produced the significant behavioral decline of the fish (Figs. 4A and 4E). Fluoroacetamide (one-way ANOVA, F (4, 55) = 8.489, $p < 0.0001$; Dunnett's T3 post hoc test, $p = 0.9538$), diphacinone (one-way ANOVA, F (4, 55) = 3.180, $p = 0.0202$; Dunnett's T3 post hoc test, $p = 0.9991$), bromadiolone (one-way ANOVA, F (4, 55) = 7.145, $p = 0.0001$; Dunnett's T3 post hoc test, $p = 0.9933$) and flocoumafen (one-way ANOVA, F (4, 55) = 3.045, $p = 0.0244$; Dunnett's T3 post hoc test, $p = 0.9770$) proved to be less toxic at the concentration of 1 mg/L, and had essentially no effect on behavior (Figs. 4B, 4D, 4G and 4H). Importantly the exposure to all the toxins at concentration of 10 mg/L led to

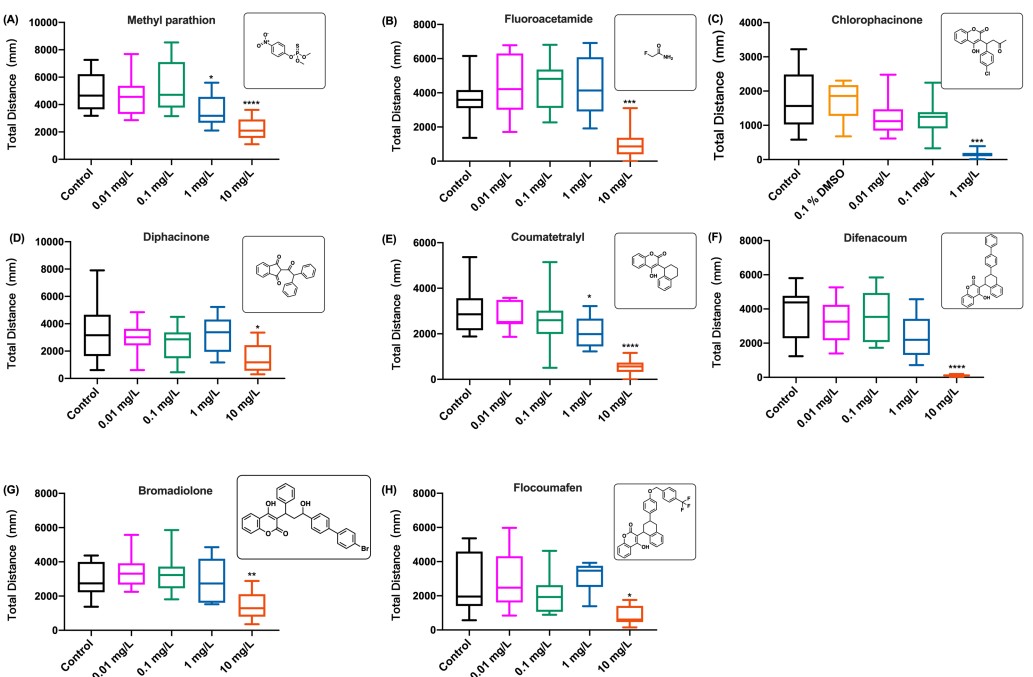

**Figure 4** **The distance travelled by zebrafish larvae during the second hour of exposure to different aquatic contaminates.** (A) Methyl parathion, (B) fluoroacetamide, (C) chlorophacinone, (D) diphcinone, (E) coumatetralyl, (F) difenacoum, (G) bromadiolone, (H) flocoumafen. Data are presented as boxplots with the median (12 larvae per treatment) and analyzed by one-way ANOVA followed by Turkey post-hoc test. * indicates the statistically significant difference compared to control. Significance was defined as *$p < 0.01$, ** $p < 0.01$, *** $p < 0.001$ and **** $p < 0.0001$.

the significant decrease in behavioral ability (chlorophacinone at 10 mg/L was lethal to larvae). For example, a significant reduction of swimming distance was revealed in fish with 10 mg/L of diphacinone (Fig. 4D) and flocoumafen (Fig. 4H) (Dunnett's T3 post hoc test: $p = 0.0104$ and 0.0485, respectively). The exposure to 10 mg/L of fluoroacetamide (Fig. 4B) resulted in significant decrease compared to the control (Dunnett's T3 post hoc test: $p = 0.0009$). The most significant differences were observed in groups of 10mg/L of methyl parathion, coumatetralyl and difenacoum compared to the control (Figs. 4A, 4E and 4F) (Dunnett's T3 post hoc test: all $p < 0.0001$). Summarized behavioral inhibition capacities ($IC_{50}$) and stabilities of toxins were provided in Table 1, which indicated that the most toxic rodenticide was chlorophacinone with an $IC_{50}$ of 0.1 mg/L. This is in the same order of magnitude as the $IC_{50}$ value determined for KCN. The least neurotoxic anticoagulant rodenticide was flocoumafen with an $IC_{50}$ value ~10 mg/L, which is two orders of magnitude higher than the $IC_{50}$ of chlorophacinone.

## Clustering of phenomics data

Clustering is a statistical technique that is important for organizing, classifying and summarizing data by grouping a set of data in a way that similar data are considered to be in the same group. In order to explore the sensitivity of zebrafish to different toxicants within 2 h, different behavioral end points, including total distance traveled, sinuosity and

Wang et al. (2021), *PeerJ*, DOI 10.7717/peerj.12703

**Table 1 Characteristics and toxicity of chemicals used in this study.** $DT_{50}$ is defined as the time it takes for an amount of a compound to be reduced by half through degradation in water (pH 7, T = 50 °C). $IC_{50}$ the concentration that reduced swimming distance of zebrafish larvae during 60–120 min exposure by 50% compared to the control group as determined in this study.

| Toxicity mechanism | Toxins | $DT_{50}$ | Residual in the water environment (ng/L$^{-1}$) | LD50[b] (rat) (mg/kg) | TDLo[b] (hunman) (mg/kg) | $IC_{50}$ (zebrafish) (mg/L) | Reference |
|---|---|---|---|---|---|---|---|
| Vitamin K antagonists (first generation) | Chlorophacinone | >1 year[a] | 87.0 | 2.100 | – | 0.1 | *Gomez-Canela, Barata & Lacorte (2014)* |
| | Coumatetralyl | >1 year[b] | 3.14–12.5 | 30 | – | 2.0 | *Gomez-Canela, Barata & Lacorte (2014)* |
| | Diphacinone | 14 days[b] | <90 | 1.500 | 0.29 | 8.7 | *Gale, Tanner & Orazio (2008)* |
| | Coumachlor | >5 days[b] | 3.63–84.2 | 187 | – | 7.5 | *Gomez-Canela & Lacorte (2016)* |
| Vitamin K antagonists (second generation) | Difenacoum | >1 year[a] | 0.86–6.55 | 0.680 | – | 1.5 | *Gomez-Canela, Barata & Lacorte (2014)* |
| | Bromadiolone | No significant degration[a] | 1.77–50.8 | 0.490 | 0.17 | 7.3 | *Gomez-Canela, Barata & Lacorte (2014)* |
| | Brodifacoum | 300 days[a] | 38.4 | 0.160 | 0.120 | 4.5 | *Gomez-Canela, Barata & Lacorte (2014)* |
| | Flocoumafen | >1 year[a] | 9.1–29.3 | 0.250 | – | 9.2 | *Gomez-Canela, Barata & Lacorte (2014)* |
| Inhibition of cytochrome C oxidase | KCN | stable under alkaline conditions[b] | 0.013–0.254 | 5 | 2.857 | 0.7 | *Abdulnabi (2020)* |
| Inhibition of acetylcholinesterase | Methyl parathion | 12 days[b] | 380–430 | 6.010 | – | 3.2 | *Hashmi et al. (2019)* |
| Disruption of the citric acid cycle | Fluoroacetamide | >2.4 yearsc | – | 5.750 | 23 | 10.0 | – |

**Notes.**
[a] Data from *Regnery et al. (2018)*.
[b] Data from the National Center for Biotechnology Information (https://pubchem.ncbi.nlm.nih.gov).
[c] Data from the Handbook of Environmental Fate and Exposure Data for Organic Chemicals (*Howard, 1989*).

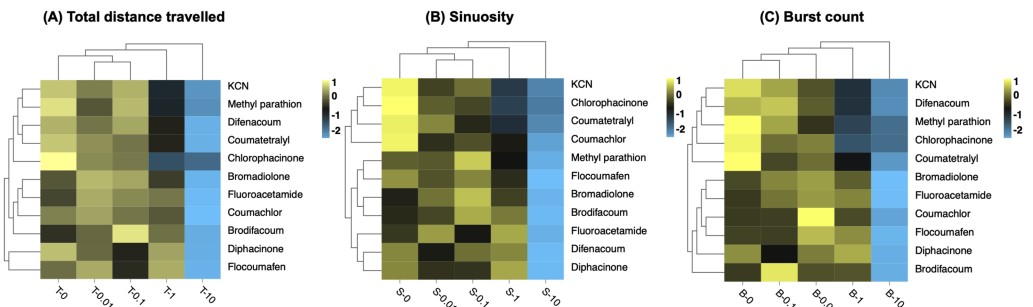

**Figure 5 Hierarchical clustering of behavioral toxicity effects induced by toxicants at different concentrations.** Clustering was performed separately using behavioral parameters of (A) total distance travelled, (B) sinuosity, and (C) burst count. Exposure concentrations of toxicants (mg/L) are indicated in the *x*-axis.

burst count, were further quantitatively analyzed by hierarchical clustering according to different concentration groups. The degree of similarity of the data between the different concentration groups indicated that the toxic substances produced similar degree of variations in behavioral parameters (Fig. 5). Clustering analysis of cumulative distance travelled by zebrafish larvae, a common method for assessing the neurotoxicity of toxicants to organisms, showed that the concentrations of 0.01 mg/L and 0.1 mg/L first clustered to form Cluster I, and then with 0 mg/L (control), forming Cluster II; then the Cluster II clustered with 1 mg/L to from Cluster III; finally, the Cluster III clustered with the 10 mg/L to from Cluster IV (Fig. 5A).

The clustering result of the effects of different concentrations of toxicants by sinuosity on larvae was consistent with that by total distance travelled (Fig. 5B). Burst count expresses the rapid movement of larvae during the experiment and is an additional measure of locomotion activity (*Hussain et al., 2020*). The sequential clustering results for the four concentrations by burst count indicated that toxic substances produced different degrees of variation in this behavioural parameter (Fig. 5C). In general, the clustering pattern of concentrations was consistent at the three behavioral endpoints, *i.e.,* consistent behavioral capacity at concentrations of 0 mg/L, 0.01 mg/L, and 0.1 mg/L, with a difference from 1 mg/L and the greatest difference from 10 mg/L.

Moreover, the results show that the 11 tested toxins were clustered in two major groups. The two first generation anticoagulants (coumatetralyl and chlorophacinone) were similar and they always clustered into one large group (Figs. 5A–5C). The same results were found for three second generation anticoagulants (bromadilone, brodifacoum and flocoumafen), illustrating the clustering reliability based on three behavioural parameters (Figs. 5A–5C). It was worth noting that clustering by sinuosity gave the best results (Fig. 5B). All the second-generation anticoagulants (bromadilone, brodifacoum, difenacoum and flocoumafen) were clustered into one large group and most first generation anticoagulants poisons (except for diphacinone) were clustered into one large group, revealing the potential of clustering by sinuosity for the analysis of unidentified poisons.

## DISCUSSION

This study is the first report on the behavioral toxicity on several highly toxic substances, including potassium cyanide, methyl parathion, fluoroacetamide, diphcinone, flocoumafen, coumachlor, chlorophacinone, coumatetralyl, difenacoum, brodifacoum and bromadiolone on zebrafish larvae. In line with the well-reported behavioral toxicity of environmental pollutants (*Chagas et al., 2019*), we observed the hypolocomotor action of these toxins in zebrafish, indicated by the reduced swimming distances. Toxins at sublethal doses (0.01, 0.1, 1 and, in some cases, 10 mg/L) evoked a dose-and time-dependent effect, markedly reducing swimming distance in zebrafish during the second hour of exposure, but not the first hour of exposure, indicating delayed toxicity. The method could be further applied in combination with metabolomics, transcriptomics, or proteomics to discriminate the mechanism of unknown toxicants.

Generally, the toxicants we tested included aerobic respiration inhibitors (*e.g.*, cyanide), acetyl cholinesterase inhibitors (*e.g.*, organophosphate pesticides), vitamin K antagonists (*e.g.*, first- and second-generation rodenticides) and organic fluorine rodenticides (*e.g.*, fluoroacetamide), which are highly toxic to humans and other mammals. For example, the rapid toxicity of cyanide is mainly due to the inhibition of cytochrome c oxidase-dependent cellular respiration, leading to the potential symptoms ranging from dizziness, headache, and hyperventilation to loss of consciousness, impaired hemodynamics, cardiac arrhythmia, cardiac arrest, and ultimately death in mammals. Relatively in zebrafish, the exposure to cyanide also caused typical toxic reactions such as cardiac bradycardia, neuronal necrosis, metabolic dysfunction and mortality (*Carbaugh et al., 2020*). The organophosphate exposure results in acetylcholine esterase (AChE) inhibition, the key molecular event leading to acute mortality, and have been confirmed in both mammalian and aquatic vertebrates models (*Schmitt et al., 2019*). Another important type of toxic pollutants are the rodenticides, including anticoagulant and organic fluorine rodenticides, causing spontaneous internal bleeding and lethality. Chlorophacinone, bromadiolone, diphcinone, brodifacoum, coumatetralyl, flocoumafen and coumachlor are first- and second-generation rodenticides, affecting vitamin K-dependent coagulation factors in the liver and causing spontaneous internal bleeding at lethal doses. Recently, an ecological risk evaluation of anticoagulant rodenticides to freshwater fishes was provided to the European Union regulatory agency by industrial manufactures (*Regnery et al., 2018*), indicating that certain fish species may be very sensitive to anticoagulant rodenticides. In addition, zebrafish were reported to be especially sensitive to waterborne exposures of anticoagulant rodenticides (*Schmitt et al., 2019*).

In the aquatic environment, zebrafish larvae are exposed to toxic substances through gills and skin, resulting in the toxicity to organs and nerves, which can be quantified by measuring behavioral capacity (*Irons et al., 2010*). Here, the effects of toxicants on the behavioral inhibition in zebrafish larvae were first assessed with KCN to optimize the experimental conditions and then the experimental approach was employed with other common toxicants (organophosphate and rodenticides). In the present study, the alternating light and dark test conditions induced cyclic changes in the swimming

distances of zebrafish, which was consistent with earlier reports (*MacPhail et al., 2009*; *Colwill & Creton, 2011*).

For the BEWS to function for the detection of a chemical, it should exert an immediate and measurable effect on fish. Here, the results of KCN exposure showed a dose dependent reduction in the swimming distance of larvae during the first and second hours of exposure. This indicated that behavioral changes of juvenile fish can be used as an effective warning sign of the presence of KCN in the aquatic environment with an $IC_{50}$ of 0.7 mg/L, which was lower than previous reports using lethality and physiological endpoints of zebrafish larvae exposed to different doses of cyanide. For example, *Nath et al. (2013)* applied zebrafish as a viable model system with the aim to find antidotes for KCN, and they found that the dose to achieve 50% reduction in heart rate in 3-dpf embryos exposed to cyanide for 2 h was 100 $\mu$M (6.5 mg/L). Dose–response effect of sublethal doses of cyanide (6.5 mg/L) on the excitation phase of the photomotor response in 30-hpf embryos was also observed after a 1-h exposure. The study also found that cyanide exposure at 500 nM (3.4 $\mu$g/L) led to significant reduction in glucose concentrations in zebrafish, although the study used about 1000 times lower KCN concentrations than the current study. Previous studies have demonstrated that zebrafish sensitivity to cyanide strongly depends on the developmental phase, which should be taken into account when designing assays for BEWS. For example, it was shown that the zebrafish embryos at 3 dpf were highly resistant to cyanide, while at 7 dpf the fish energy metabolism became sensitive to potassium cyanide exposure at 25 $\mu$M (1.6 mg/L) (*Sips et al., 2018*). A different study (*Carbaugh et al., 2020*) showed a significant concentration- and developmental stage-specific effects on the photomotor response of zebrafish embryos (24 hpf or 120 hpf) following sodium cyanide exposure at 33.3 $\mu$M (1.6 mg/L), which could be applied in the rapid screening test for neurobehavioral effects.

Here, the optimized test conditions using KCN were used to assess the behavioral impacts of other organic pollutants. The organophosphorus and anticoagulant rodenticides all resulted in significant behavioral depression in zebrafish larvae within the concentration range of 0.01–10 mg/L, which was consistent with the previous reports. For example, *Jin et al. (2013)* reported that zebrafish larvae at 8 dpf exposed to organophosphorus for 16 h exhibited several phenotypes similar to the human response to organophosphorus, including behavioral deficits, paralysis, and eventual death. The semi-lethal values for methylthion and parathion were 54 $\mu$M (17.1 mg/L) and 13.5 $\mu$M (3.9 mg/L), respectively. They demonstrated the toxicity related to acetylcholinesterase inhibition by liquid chromatography/tandem mass spectrometry-based metabolite analysis. The acute toxicity of anticoagulant compounds to non-target organisms such as zebrafish has been poorly reported, with a major focus on warfarin. Exposure to warfarin for 2.5 days increased mortality in zebrafish in a dose- and time-dependent manner and appeared to be more toxic during embryonic development (*Granadeiro et al., 2019*). The highest dose of warfarin (125 mg/L) is capable of causing hemorrhage, shortened lifespan, reduced growth, shortened endochondral bone length and delayed mineralization of skeletal structures, with toxicity involving the redox system, blood coagulation and angiogenesis, visual phototransduction and collagen formation.

Variations in the structure and molecular polarity of toxicants can result in different rates of uptake through the skin and gills of zebrafish larvae and also in different mechanisms of neurotoxicity. In the current study, this was reflected by the variation in the time of onset of the behavioral changes in zebrafish larvae as well as variation in the chemical doses which induced effects. The dose-dependent effects were not observed for coumachlor and brodifacoum within the 1-h exposure time. This might have been caused by inadequate exposure time and different rate of absorption/metabolism of the chemical. The results herein showed that the presence of coumachlor in the water during the first hour of exposure resulted in increased swimming activity of larvae (Fig. 2B), which was consistent with reports of flight behavior to avoid the exposure to the pollutants (*Magalhaes Dde et al., 2007*). In addition, we also observed that there was no significant difference in the behavioral capacity within 1 h of exposure to brodifacoum exposure (Fig. 2C), indicating that both the concentration of the toxicant and the duration of exposure were within the no observed adverse effect concentration (NOAEC) values of the agent (*Magalhaes Dde et al., 2007*). The lack of significant differences or the appearance of abnormal responses during the short exposure time of 1 h might be explained by the adaptation of the subject during the experiment. However, the organisms would not maintain a chronic state of stress response. Our results showed that most of the compounds tested here exhibited significant behavioral toxicity to zebrafish at 6 dpf during the second hour of exposure, and that the behavioral toxicity of chemicals with different properties was dose-dependent over the 60 to 120 min assessed. In our study, several toxins produced significant behavioral changes at concentrations of 1 mg/L or 10 mg/L, and chlorophacinone was the most toxic compound based on the behavioral toxicity, with an $IC_{50}$ of 0.2 mg/L. These observations were consistent with previous reports where behavioral toxicity was employed as a biological warning system, and showed high sensitivity at sublethal concentrations of heavy metal pollutants (*Nusser et al., 2016*; *Li et al., 2019*) and pesticides (*Hussain et al., 2020*).

Among the several behavioral parameters of zebrafish larvae, locomotion is probably one of the biologically most important traits, which has been measured for toxicity assessment in pharmacology, neurobiology, genetics, and ethology (*Ingebretson & Masino, 2013*). In our study, toxicity was determined by calculating significant differences in the distances travelled without the need for complex and expensive equipment or transformation of the data (*Oliva Teles et al., 2015*). Several groups have reported high-throughput behavioral data for zebrafish larvae. For example, *Nusser et al. (2016)* proposed a new parameter for evaluating zebrafish stress upon acute exposure to sublethal doses of $CdCl_2$ and permethrin –an endpoint that allowed for the distinction between the overall movement and directed movement of larvae to describe the avoidance behavior of zebrafish at 4 dpf. In our study, three behavioral endpoints, namely, total distance traveled, sinuosity and burst counts, were selected for the evaluation of behavioral capacity. The first endpoint describes the total swimming activity of zebrafish during exposure to toxicants, the second describes the ability of zebrafish larvae to balance with reduced Parkinsonian-like behavior similar to that caused by neural damage (*Bortolotto et al., 2014*) and the third describes rapid movement during the experiment, which helps to evaluate the locomotion activity of zebrafish larvae (*Hussain et al., 2020*). Importantly, all of the highly toxic substances induced inhibition of all

three behavioral endpoints in zebrafish in the tested concentration range (0.01–10 mg/L), demonstrating the feasibility of toxicant warning through the inhibition of behavioral capacity. The method we established has limitations for the early detection of different toxins, mainly because most toxins can only cause behavioral changes in zebrafish larvae at high concentrations (mg/L level). However, we believe it is still important to develop rapid (within two hours) biological tests, which could provide timely warning of sudden water contamination and drinking water poisoning.

## CONCLUSIONS

In this work, we assessed changes in the zebrafish larval behavior (swimming distance and associated parameters) upon exposure to cyanide, pesticides and rodenticides at sub-lethal concentrations to propose a BEWS for these chemicals. Our results suggest that the method employed herein successfully indicated water contamination of diverse types of chemicals. After exposure to toxins, such as potassium cyanide, methyl parathion, fluoroacetamide, diphcinone, flocoumafen, coumachlor, chlorophacinone, coumatetralyl, difenacoum, brodifacoum and bromadiolone (1 or 10 mg/L), significant reductions in zebrafish larvae travel distance were observed (ANOVA $P < 0.05$), indicating detrimental effects on the larval behavior during the 2 h experimental period. Although, some studies have used adult or juvenile zebrafish to detect water contamination, to our knowledge, this is the first time that detection for early warning of rodenticides in water contamination has been achieved through behavioral changes in zebrafish larvae. In addition, the BEWS method developed in this study is rapid, reliable, versatile and adaptable to a wide range of highly toxic substances. Taking advantage of the sensitivity, robustness, high throughput and low ethical risk of larval zebrafish, we anticipate that this model will prove useful for risk assessment of various toxicants.

### Funding
This work was supported by Zhejiang Province Key Research and Development Program (No. 2021C03135), National Key Research and Development Program of China (No. 2018YFC0807201), grants from Basic Public Welfare Research Program of Zhejiang Province (No. LGF20C090001), Hangzhou Agricultural and Social Development Research Initiative design project (No. 20190101A08) and Zhejiang Police University Cooperative Scientific Research Project (No. 2019XJY002) and National College Students Innovation and Entrepreneurship Training Project (No. 202011483013). The funders had no role in study design, data collection and analysis, decision to publish, or preparation of the manuscript.

### Grant Disclosures
The following grant information was disclosed by the authors:
Zhejiang Province Key Research and Development Program: No. 2021C03135.

National Key Research and Development Program of China: No. 2018YFC0807201.
Basic Public Welfare Research Program of Zhejiang Province: No. LGF20C090001.
Hangzhou Agricultural and Social Development Research Initiative design project: No. 20190101A08.
Zhejiang Police University Cooperative Scientific Research Project: No. 2019XJY002.
National College Students Innovation and Entrepreneurship Training Project: No. 202011483013.

## Competing Interests

The authors declare there are no competing interests.

## Author Contributions

- Binjie Wang conceived and designed the experiments, analyzed the data, prepared figures and/or tables, and approved the final draft.
- Junhao Zhu and Anli Wang performed the experiments, prepared figures and/or tables, and approved the final draft.
- Jiye Wang analyzed the data, authored or reviewed drafts of the paper, and approved the final draft.
- Yuanzhao Wu and Weixuan Yao conceived and designed the experiments, authored or reviewed drafts of the paper, and approved the final draft.

## Animal Ethics

The following information was supplied relating to ethical approvals (i.e., approving body and any reference numbers):

The zebrafish experiments were approved by the Zhejiang University Experimental Animal Welfare and Ethical Review Committee.

## Data Availability

Total distance travelled, sinuosity, and burst count were used for statistical analyses to compare significant differences in behavioural changes in juvenile zebrafish at different concentrations and are available in the Supplemental File.

## Supplemental Information

Supplemental information for this article can be found online at http://dx.doi.org/10.7717/peerj.12703#supplemental-information.

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
