# Peer review of "Early detection of cyanide, organophosphate and rodenticide pollution based on locomotor activity of zebrafish larvae"

_PeerJ, doi:10.7717/peerj.12703_

## Round 0.1 · original submission · Major Revisions

The reviewer comments are clear, so I will not repeat them here. In general, most of the comments reveal concerns about the sensitivity of the system since the concentrations used are several orders of magnitude higher than is commonly found in the environment. Please address this and other concerns in your revision.

Reviewer 1 ·

Excellent Review

This review has been rated excellent by staff (in the top 15% of reviews)
EDITOR COMMENT
The reviewer conducted a thorough analysis of the manuscript and provided constructive comments focused on the science.

Basic reporting

General comment: My main concern in this section is that authors propose the developed system “for utilization as part of a monitoring system and an ideal tool for environmental risk assessment”. In the introduction information of environmental relevant concentrations of all these chemicals in aquatic ecosystems, both non-directly exposed and directly exposed, is currently missing. This is basic information to understand the sensitivity of the proposed system. It is not my role to look for this info in the bibliography, but I think that the concentrations used for the authors are several orders of magnitude higher than the commonly found in the environment. I would be very surprised if rodenticides were found in aquatic ecosystems not directly exposed at concentrations higher than low ng/L… If this is the case, the proposed system is not sensitive enough for environmental applications (at least for rodenticides…)

Specific comments:

Lines 41-43: “Toxic substances, such as cyanide, organophosphates and rodenticides, have appeared in the water environment, and have even been reported to cause poisonings in humans and public safety emergencies”. I agree with the authors that: (1) it is possible to find these chemicals in water, and (2) that there are cases in humans of poisoning for most of them. But these (1) and (2) are two independent facts. In this sentence, reader can understand that levels of these compounds in water can result in human poisoning. And this is not what the references say (all of them are about suicides, homicides...). Therefore, the next concern raising is about the toxicity or safety of the normal range of concentrations of these chemicals in aquatic ecosystems, that has not been indicated in the manuscript. Authors should provide detail on the environmental levels of the eleven chemicals they have been testing in the manuscript.

Lines 60-62: “Importantly, the existing studies correlated behavioral changes of zebrafish induced by toxicant exposure with alterations in physiological indicators… “ Please, provide references.

Lines 86-88: “Compared with adult zebrafish, the behavioral changes of larvae occur at much lower toxicant concentrations, allowing earlier detection of contamination” I can't agree with this statement. It depends on the toxicant. For instance, for OP insecticides embryos or early larvae are more resistant than the adults, as adults die very quickly after respiratory arrest. As in embryos and early larvae an important part of the gas interchange takes place through the skin, they are able to increase the survival.

Lines 93-94: “In another study, the exposure to low concentrations of pesticides (1 ppb) in the aquatic environment…” In my opinion 1 ppb it is not necessary a low concentration. Many pesticides are commonly found in aquatic ecosystems in the range of low ng/L

Lines 104-105: “The aim of this work was to develop a robust system with sensitive detection capability of several toxicants in water, based on the behavioral changes of zebrafish larvae” It seems that authors intend to develop a sensitive system for detecting some chemicals in water. But first of all they should provide information on the range of concentrations of each chemicals in aquatic ecosystems, both in directly exposed locations and in non-directly exposed locations. If they are able to detect these chemicals at concentrations in the range of non-directly exposed ecosystems, then they'll have developed a sensitive system suitable for environmental monitoring. However, this hasn’t been demonstrated by the moment.

Experimental design

General comment: I have a problem with the behavioral analysis performed in this study. It has sense to analyze behavior when there is no signs of system toxicity, but it seems that only lethality was recorded. There is no indication on the general state of the larvae at the highest tested concentrations. We can see a significant decrease at the highest concentrations only because the larvae were near the dead or it was a specific effect on the nervous system?

Line 120: “The adult wildtype zebrafish (~3 months old, 2.5–3.0 cm in length)“. I must to say that I’m a bit surprised by the fact that authors have been able to obtain 3 months zebrafish with 2.5-3.0 cm. Usually it takes at least 1 year to get this size.

Lines 124-125: “The selected fish eggs were checked daily, and dead and malformed embryos…” Once fertilized, the oocyte becomes an embryo. Please, change the term “eggs” by “embryos”.

Lines 125-126: “Because the embryo receives nourishment from yolk sac, no feeding was required during the experiment”. At 28ºC most of the yolk is consumed during the first 5 days.

Lines 128-129: “At the end of the experiment, the larvae were euthanized with a saturated benzocaine solution and stored at -78º C” In the manuscript only behavioral endpoints are analyzed. Why are then stored the larvae? The true is that to have some additional endpoints (transcriptomics or metabolomics) to integrate with the behavioral endpoints should be great. The point is that if some additional analyses are going to be done with the exposed larvae, the best is to join all the info in one manuscript.

Lines 138-140: “The stock solutions of KCN, difenacoum, bromadiolone, diphcinone, coumatetralyl, flocoumafen, and fluoroacetamide were prepared in deionized H2O and stored in the dark at 4 °C” For how long the stock solutions were stored at 4ºC? Have you any evidence that there was no degradation during the stored time?

Line 141: “DMSO in the exposure solutions did not exceed 0.1%” I'm not sure if I understand correctly, but authors should explain if the final DMSO concentrations in the working solutions were different with the different concentrations of the same chemicals or among chemicals...

Line 144: In the section “Chemical exposure” should be indicated the exposure time. For me is unclear: 60 min exposure and then transferred to clean water for behavioral analysis? 60 min exposure and then behavioral analysis with the larvae still exposed? Please, clarify.

Lines 161-162: “The activity of each zebrafish larvae was analyzed during each 60 min period for three endpoints, namely, total distance traveled (mm), sinuosity (°/s) and burst count…” Please, clarify the meaning of "each 60 min period". It means the summary of the 6 periods of light (10 min each) and the 6 period of dark (10 min each)?

Lines 167-169: “The inhibitory capacity of the toxicant on the behaviour of zebrafish was assessed as the IC50, i.e. the concentration at which the cumulative distance travelled was reduced by 50% over 60-120 min compared to the control group”. I only want to congratulate the authors because they are very lucky. They have found the IC50 for 11 chemicals within the same range of concentrations (10 ppb-10 ppm) !!!

Line 170: I have seen that authors have tested normality and homocedasticity of the different distributions and they have used then the appropriate statistical test. This is a good point. However, this raise a new question: if there are some data that don’t follow a ormal distribution, why all the graphs in the Figures are based in parametric statistics (mean and SEM)? In my opinion it would be better to use boxplots with the median instead of the current bar charts.

Validity of the findings

Lines 185-186: “The larval zebrafish (6 dpf) were exposed to different concentrations of KCN (0, 0.01, 0.1, 1, 10 mg/L) for 2 h and their locomotor activity was monitored” It seems that larvae were exposed for 2h, but this is also the time covered by the behavioral analysis. I'm not sure if behavioral analysis started at the end of the 2h exposure. I don't know if larvae were also exposed during the 2 h of the behavioral analysis. All these details should be clarified.

Lines 191-194: I don't understand all the effects that authors claim to see: inhibition, excitation, avoidance... As I can't see any statistics after each comment, I suppose that there are no statistically significant differences, and therefore, there are no effects.

Lines 197-200: “Then, we assessed the behavioral impairment of zebrafish using the cumulative distances travelled by zebrafish larvae over the three cycles totaling the first 60 min (0-60 min) and over three cycles of the second 60 min (60-120 min) of exposure” I'm not extremely happy with this decision from authors. First of all, authors should justify the need to perform 6 cycles L/D. Why they are performing 6 cycles? What information they expect to obtain performing 6 cycles instead of 1 cycle? From my point of view, it is not very interesting to pooling the locomotor activity of L1/D1/L2/D2/L3/D3 and L4/D4/L5/D5/L6/D6. I don't know what exactly authors intend to obtain.

Lines 205-206: “Since 10 mg/L KCN was apparently lethal, there was no swimming detected in this exposure group” I don't understand the meaning of "10 mg/L KCN was apparently lethal" All the larvae died? Some larvae died? Not only lethality, but also any sign of system toxicity should be carefully surveyed before to start any behavioral test. On the other hand, why authors don’t include here the description of the effects on sinuosity and burst behaviors??

Lines 208 and 226: The same section title!

Lines 232 and 239: Exactly the same statistical information is provided.

Lines 287-288: “Clustering analysis of cumulative distance travelled by zebrafish larvae, a common method for assessing the neurotoxicity of toxicants to organisms” I'm not so sure that accumulative distance across 3 different L/D cycles is to common method, as it is unclear usefulness of the information that this approach provides.

Lines 292-294: “The reduction of sinuosity on zebrafish larvae indicated that the balance of movement was disrupted, and has been reported in Parkinson's-like behavior resulting from nerve injury (Bortolotto et al., 2014)” This sentence is not related with the results, but with the discussion.

Lines 299-302: “In general, the clustering pattern of concentrations was consistent at the three behavioral endpoints, i.e., consistent behavioral capacity at concentrations of 0 mg/L, 0.01 mg/L, and 0.1 mg/L, with a difference from 1 mg/L and the greatest difference from 10 mg/L” It was really necessary to perform the hierarchical clustering to get this conclusion?

Reviewer 2 ·

Basic reporting

1- The manuscript entitled “Early detection of cyanide, organophosphate and rodenticide pollution based on locomotor activity of zebrafish larvae” focus a relevant topic in ecotoxicology, namely by using a behavior approach on zebrafish model to detect early responses to contaminants. Despite the high relevance and applicability of this approach, I highlight some issues related with the methodological approach, which might compromise the interpretation of results and outcomes of this manuscript.
2- The objectives of the work are specified but the hypothesis tested is not presented.
3- The introduction is well structured but the authors do not present the rationale behind the selection of the contaminants for this behavioral study. The authors present the mode of action in target species for the selected contaminants, but an explanation on why they were selected is missing. Why do they specifically fit the purpose of testing this behavioral assay to detect contaminants?
4- The results presented address the main objectives of the work and are well structured and analysed.
5- Control in each figure: specify if it is relative to negative control or to solvent control. Present data on DMSO effects on behavior relatively to negative control.

Experimental design

1- The final concentration of DMSO used in all treatments (0.1%) is above the maximum concentration recommended for solvent use in standard guidelines, including in FET OECD 236 Guideline: “Where a solvent is used to assist in stock solution preparation, its final concentration should not exceed 100 μl/L and should be the same in all test vessels.” This corresponds to a maximum percentage of 0.01% (v/v) of solvent. As solvents are well known to cause effects in behavior, the solvents should be kept at minimal concentrations, and whenever possible, well below recommended values in order to avoid biased results and/or mixture effects from solvent + target substance.
2- Chemical analysis is lacking in the methodological approach. The inclusion of this data is extremely relevant to fit the purpose and objective of testing the behavioral assay to detect the selected substances. In order to assure that the effects observed in the fish are elicited by a certain substance concentrations, the real concentrations tested should be properly verified to check the nominal concentrations used. The authors should add analytical chemical analysis data (stock solutions, initial solutions and) to this work to properly interpret their results and outcomes.
3- Line 172-173 – “The adaptation period included in the analysis was an initial 10-min dark period”. This is already explained in the paragraph before (lines 155-157). However, if presented again here in statistical analysis, the authors should state if they have included in the data analysis the acclimation period or not. In fact, the acclimation period should not be included in the data analysis and this mention here is confusing.

Validity of the findings

1- The discussion should be more focused on interpreting and discussing the specific results obtained instead of giving a general overview about the topic. This will better fit the Introduction section. In general, the discussion lacks interpretation of the specific results obtained for the chemicals used vs. results obtained by other authors.

Additional comments

1- Line 115-116 - Delete: “Add your introduction here”.
2- Lines 145-146 – It is not necessary to include along the manuscript text the authors’ credits and participation in the lab work. This information should be presented in a proper and different section of the manuscript according to journal guidelines.

---

## Round 0.2 · accepted · Accept

Thank you for your thoughtful and thorough responses to the reviewer comments.